# Evaluation of Diet and Symptom Severity in Disorder of Gut–Brain Interaction

**DOI:** 10.3390/jcm13144132

**Published:** 2024-07-15

**Authors:** Wioleta Faruga-Lewicka, Agnieszka Bielaszka, Wiktoria Staśkiewicz-Bartecka, Sabina Opiołka, Agata Kiciak, Marek Kardas

**Affiliations:** Department of Food Technology and Quality Evaluation, Department of Dietetics, Faculty of Public Health in Bytom, Medical University of Silesia in Katowice, 41-808 Zabrze, Poland; abielaszka@sum.edu.pl (A.B.); wstaskiewicz@sum.edu.pl (W.S.-B.); s75947@365.sum.edu.pl (S.O.); akiciak@sum.edu.pl (A.K.); mkardas@sum.edu.pl (M.K.)

**Keywords:** gastrointestinal symptoms, digestive diseases, diet, nutrition

## Abstract

**Background**: Disorders of gut–brain interaction are chronic or recurrent symptoms originating in the gastrointestinal tract that cannot be substantiated by the results of standard clinical tests, such as radiologic studies, morphologies, or endoscopic examination. The diagnosis of these disorders is mainly based on symptoms and the standardized Rome IV criteria. These criteria classify functional disorders of the gastrointestinal tract according to anatomical location and define each disorder according to a set of symptoms. **Methods**: This study was conducted between October 2021 and February 2022. Participants in the study were patients of a gastroenterology outpatient clinic with a functional disease diagnosed by a gastroenterologist. A questionnaire was used to conduct the study, with questions regarding perceived functional discomforts of the gastrointestinal tract, dietary changes to alleviate discomforts, and frequency of consumption of various food groups. **Results**: Based on the study, statistical significance was demonstrated between the gender of the respondents and the severity of gastrointestinal complaints after the consumption of legumes and alcohol. The analysis performed confirmed the correlation between the age of the respondents and the severity of complaints when consuming raw vegetables and fruits, brassica vegetables, legumes, fried products, and spicy products. There was also a significant correlation between the body mass index (BMI) of the respondents and the severity of complaints after alcohol consumption. **Conclusions**: The results identify abdominal pain, bloating, and constipation as the most commonly reported gastrointestinal symptoms among participants. The association between the consumption of certain foods, such as milk and dairy products, as well as fried and fatty foods, and the severity of disorders of gut–brain interaction symptoms was confirmed. Despite this, the majority of respondents did not eliminate any food products to alleviate the discomfort.

## 1. Introduction

Disorders of gut–brain interaction (DGBI) are chronic or recurrent symptoms originating in the gastrointestinal tract that cannot be substantiated by the results of standard clinical tests, such as radiologic studies, morphologies, or endoscopic examination [1,2]. The diagnosis of these disorders is mainly based on symptoms and the standardized Rome IV criteria. These criteria classify functional disorders of the gastrointestinal tract according to anatomical location and define each disorder according to a set of symptoms [3]. Many symptoms originating from the gastrointestinal tract, including abdominal pain, bloating, a feeling of fullness in the abdominal cavity, nausea, vomiting, diarrhea, or constipation, are highly prevalent in the community. Moreover, these symptoms are often common to many disease entities [4]. Similar complaints can occur during the presence of gastrointestinal cancer, inflammatory bowel disease, or gastrointestinal motility disorders. However, in the case of a significantly large number of patients presenting to gastroenterology outpatient clinics, studies do not identify underlying structural abnormalities that would explain the previously mentioned symptoms. Therefore, they are classified as “functional” [1,2,3,4].

A worldwide study conducted by the Rome Foundation found that more than 40% of people worldwide suffer from DGBI, which affects the comfort of their lives, as well as the healthcare aspect [5]. In addition, it has been shown that women are far more likely to suffer from DGBI than men. Functional dyspepsia, irritable bowel syndrome (IBS), constipation, and functional bloating are among the most common disorders among the global population [5]. Over the past few years, some mechanisms have been documented that may play key roles in the pathophysiology of these disorders, namely, dietary factors, infections, genetic factors, disturbances in the intestinal microbiome, mucosal inflammation, and many others. Despite this, functional disorders of the gastrointestinal tract are still insufficiently understood, due to their complex and multifactorial nature [6,7].

Functional gastrointestinal disorders have long been categorized using the Rome criteria, which help systematize symptoms often not linked to organic changes. The Rome IV criteria, updated in 2016, redefine these disorders as gut–brain interaction disorders. They classify disorders into six main categories based on the gastrointestinal tract segment, specifically tailored for adults, children, and adolescents [8,9,10].

The pathogenesis of disorders caused by inappropriate gut–brain interaction involves changes in the bidirectional interactions between the brain and gut, which are implicated in the pathogenesis of irritable bowel syndrome (IBS) and related functional gastrointestinal disorders. The gut microbiota and its metabolic products can influence gut functions such as intestinal permeability, mucosal immune function, and the hypothalamic–pituitary–adrenal (HPA) axis activity. Dysbiosis, or the imbalance of the gut microbiota, may contribute to IBS by increasing intestinal permeability, activating the mucosal immune response, enhancing visceral sensitivity, and altering gut motility [11,12,13].

Patients with chronic abdominal pain and IBS show differences in the organization of the gut microbiota and the production of its metabolites. The benefits of manipulating the microbiome with prebiotics, probiotics, or antibiotics in IBS patients remain controversial. It is unclear whether observed changes in the microbiome of IBS patients result from altered gut function, physiology, or brain signaling. Mechanisms such as the hypothalamic–pituitary axis, the autonomic nervous system (ANS), and ANS modulation of the enteric nervous system can influence the gut microbiota environment. These mechanisms can alter motility patterns in different regions of the intestine, epithelial permeability, luminal secretion, mucosal immune function, and neurotransmitter release [14,15].

Some correlation is apparent between food preferences and choices and the severity of gastrointestinal complaints. Individuals who choose foods high in fat, fried, spicy products/foods, or who consume fast food are far more likely to experience gastrointestinal discomfort compared to those who follow a healthy diet [16,17]. Ingested foods have an inherent impact on visceral sensitivity, intestinal barrier function, gastrointestinal motility, or the microbiome. In the context of IBS, the relationships between the effects of milk and dairy intake, fiber-rich foods, carbohydrates, and food additives and sweeteners on triggering symptoms are most commonly studied [16]. On the other hand, among patients with functional dyspepsia, symptoms such as bloating and rapid satiety are often identified with the type of food consumed. A disadvantageous tendency between the consumption of wheat, fat, and alcohol, as well as coffee, tea, sodas, fruit juices, and fruits and milk is also looked for [18]. Triggers of symptoms within those struggling with functional nausea, heartburn, belching, and vomiting are primarily foods that stimulate gastric secretion. This applies mainly to hot spices (bell pepper, hot peppers, chili), citrus, carbonated beverages, acidic foods and products, and fruit juices. Equally detrimental to increased gastric acid secretion are foods with a high-fat content, alcohol, coffee, strong teas, essential and aromatic decoctions of both vegetables and meat, as well as chocolate products [19].

The purpose of this study was to evaluate the diets of people with functional gastrointestinal disorders, investigate the frequency of gastrointestinal complaints, assess the impact of food products on the severity of complaints, and identify the most frequently excluded foods for gastrointestinal function problems. It was hypothesized that people with functional gastrointestinal disorders adapt specific dietary patterns that affect the frequency and severity of gastrointestinal complaints. Food items may have a direct impact on the intensity of symptoms, resulting in the frequent exclusion of certain foods from the diet. It is also hypothesized that there is a set of products that are most often eliminated by this population to alleviate symptoms.

## 2. Materials and Methods

### 2.1. Study Design

This study was conducted between October 2021 and February 2022. Participants in the study were patients of a gastroenterology outpatient clinic with a functional disease diagnosed by a gastroenterologist. Participants who consented to take part in the study completed questionnaires after their outpatient clinic visit. The response rate was 90.54%. A questionnaire was used to conduct the study, with questions regarding perceived functional discomforts of the gastrointestinal tract, dietary changes to alleviate discomforts, and frequency of consumption of various food groups.

All study participants were informed about the purpose of the study and its anonymity, and were asked to accept the rules of data sharing. Information about informed and voluntary participation in the study was provided at the beginning of the questionnaire. The Declaration of Helsinki of the World Medical Association guided the conduct of this study. This study was approved by the Bioethics Committee of the Silesian Medical University in Katowice (PCN/0022/KB/299/19/20, date of approval: 29 January 2020) in light of the Law of 5 December 1996, in the Profession of Physician and Dentist, which includes a definition of medical experimentation.

### 2.2. Research Tools

A survey questionnaire was used to conduct the study, which consisted of a metric (subject’s data: age, education, and anthropometric data—height and weight), the QEB questionnaire (Questionnaire of Eating Behavior, Questionnaire for the Study of Eating Behavior and Opinions on Food and Nutrition) developed by the Behavioral Determinants of Food Team, PAS [20], and the author’s section on the prevalence of functional gastrointestinal complaints. Appendix A includes a survey questionnaire.

For the QEB questionnaire, questions on the subjects’ dietary habits (questions 1–8, 11, 13, 14, 20, 21, 27–29, 34–37, 42) and frequency of food intake (questions 9, 10, 12, 15–19, 22–26, 30–33, 38–41) before and during pregnancy were used. For questions on the frequency of consumption of various foods, a ranking method was used, where the answer “never” received a rank of 1, “1–3 times a month” a rank of 2, “once a week” 3, “several times a week” 4, “once a day” rank 5, and “several times a day” rank 6.

The author’s portion of the questionnaire addressed the frequency of complaints such as abdominal pain, nausea, constipation, bloating, belching, diarrhea, excessive gas, overflow, chronic fatigue, and weakness. To help identify symptoms, the survey included an explanation of DGBI-related terms. “Abdominal pain” meant pain in the area between the bottom of ribs and pelvis. “Nausea” meant a feeling of sickness with an inclination to vomit. “Constipation” meant a condition in which a respondent has uncomfortable or infrequent bowel movements. “Bloating” meant a condition where the abdomen feels full and tight, often due to gas. “Belching” meant emission of wind from the stomach through the mouth. “Diarrhea” meant loose and watery stools. “Excessive gas” meant farting more than 25 times per day. “Overflow” meant sounds coming from the abdominal cavity, which are caused by the movement of the intestines as they move food. “Chronic fatigue” meant prolonged feeling of tiredness. “Weakness” meant lack of strength, firmness.

The responses were ranked using the terms “frequently” (meaning more than 3 times a week), “occasionally” (meaning several times a month), and “never”. Subsequent questions were asked about the severity of discomfort through the consumption of specific food groups, such as milk and dairy products, raw vegetables and fruits, legumes, coffee, alcohol, fried foods, and spicy foods.

The subjects’ nutritional statuses were assessed using body mass index, calculated using the following formula: BMI (kg/m^2^) = body weight (kg)/height (m)^2^. Subsequently, the results were interpreted according to the World Health Organization (WHO) guidelines, where a BMI value ≥ 30.00 kg/m^2^ indicates obesity, values of 25.00–29.99 kg/m^2^—overweight, 18.50–24.99 kg/m^2^ indicates normal weight, values of 17.00–18.49 kg/m^2^ are underweight, and BMI ≤ 16.99 kg/m^2^ indicates malnutrition [21].

### 2.3. Study Group

The study group consisted of 234 people, of whom 70.1% (*n* = 164) were women and 29.9% (*n* = 70) were men. By age of respondents, the largest group was those under 30 years of age (32.5%, *n* = 76), followed by 31–40 years of age (31.6%, *n* = 74), 41–50 years of age (27.8%, *n* = 65), and the smallest group was those over 51 years of age (8.1%, *n* = 19). The average age of respondents was 36.15 ± 7.6.

Among the respondents, 47% of them reported having higher education (*n* = 110), 44.9% (*n* = 105) had secondary education, and 8.1% (*n* = 19) had primary education.

Inclusion criteria for the study encompassed all patients diagnosed with gastrointestinal dysfunction by a gastrointestinal physician at the clinic who qualified for the study, who consented to participate, were of legal adult age, and correctly filled out the study questionnaire. Exclusion criteria involved the presence of incorrectly completed questionnaires, comorbidities that could affect gastrointestinal function, and any food allergies or intolerances.

### 2.4. Statistical Analysis

Statistical analysis was performed using Statistica 13.3 software (TIBCO Software Inc., Palo Alto, CA, USA). Both parametric tests, which are used when assumptions about the normality of the data distribution are met, and nonparametric tests, which are used when these assumptions are not met, were used to analyze the data. The normality of continuous variables was assessed using the Shapiro–Wilk test. The values of measurable parameters (e.g., measurement results) were presented using the arithmetic average and standard deviation. Nonmeasurable parameters (e.g., qualitative scale scores) were presented using percentages. The chi^2^ test of homogeneity was used to examine differences between groups. For comparisons involving categorical variables with small sample sizes, Fisher’s exact test was used instead of the chi^2^ test to ensure accurate results. 

Cramer’s V correlation coefficient was used to analyze the correlation between gender, age, and nutritional status interpreted by assessing BMI on the severity of gastrointestinal complaints.

A value of *p* < 0.05 was taken as the level of statistical significance.

## 3. Results

### 3.1. Analysis of Perceived Gastrointestinal Complaints

Analysis of the question on perceived discomfort showed that respondents most often complained of abdominal pain, which occurred frequently in 48.3% (*n* = 113) of people, and occasionally in 37.2% (*n* = 87), and only in 14.5% (*n* = 34) of people did abdominal pain not occur. Respondents also frequently suffered from bloating, which occurred frequently in 46.8% (*n* = 109) of respondents, occasionally in 31.8% (*n* = 74), and only in 21.5% (*n* = 50) of people did it not occur. In addition, constipation occurred frequently in a similar number of respondents 47.4% (*n* = 111), sporadically in 30.3% (*n* = 71), and did not occur in 22.2% (*n* = 52) of respondents. Bouncing was absent in 43.6% (*n* = 102), while it occurred occasionally in 29.5% (*n* = 69) and frequently in 26.9% (*n* = 63). Excessive gas output affected 32.1% (*n* = 75) of individuals occasionally, frequently for 21.4% (*n* = 50) of respondents, and did not occur in 46.6% (*n* = 109) of respondents.

Respondents were next asked about changes in diet as a result of their accompanying ailments. No changes in diet were declared by 48.7% (*n* = 114) of respondents, while changes were made by 43.6% (*n* = 102) of respondents.

The negative effect of increasing gastrointestinal complaints of milk and dairy products was indicated by 38.9% (*n* = 91) of people, and the answer “sometimes” by 35.5% (*n* = 83). Another question concerned fried products. Their negative impact was indicated by 46.2% (*n* = 108) of respondents, and 29.1% (*n* = 68) sometimes. Similarly, high-fat products exacerbated discomfort in 39.7% (*n* = 93) of respondents regularly, and in 29.1% (*n* = 68) only sometimes. In contrast, alcohol (67.5%, *n* = 158), raw fruits and vegetables (58.1%, *n* = 136), and brassica vegetables (55.6%, *n* = 130) did not cause gastrointestinal discomfort in the vast majority.

### 3.2. Analysis of the Frequency of Consumption of Individual Food Groups

Respondents mostly declared that they eat from four to five meals (49.6%, *n* = 116), followed by 31.2% (*n* = 73) of respondents with only three meals. More than five meals were indicated by 14.1% (*n* = 33), and fewer than three meals by 5.1% (*n* = 12) of respondents. A total of 95 (40.6%) of the respondents confirmed that the breaks between main meals are between 2 and 3 h, while more than 3 h was indicated by 26.5% (*n* = 62) of people. On the other hand, 26.1% (*n* = 61) of subjects declared that breaks between meals are from 1 to 2 h.

The frequency of consumption of different food groups among respondents is summarized in Table 1.

Table 2 shows the results of the correlation between the respondents’ gender, age, and BMI value and the occurrence of selected gastrointestinal ailments.

The statistical analysis conducted showed a correlation between the gender of the respondents and the occurrence of chronic fatigue, weakness, and overflow. A correlation was shown between the age of the respondents and the occurrence of such complaints as bloating, fasting abdominal pain, and chronic fatigue and weakness. In addition, a correlation was noted between BMI and the problem with the occurrence of nausea, chronic fatigue and weakness, and nocturnal abdominal pain.

In addition, a correlation analysis was performed between the effects of gender, age, and nutritional status, interpreted by assessing BMI on the severity of gastrointestinal complaints. There was no statistically significant correlation between gender (*p* = 0.23; V = −0.231) and age (*p* = 0.54; V = 0.185) and presenting complaints. There was a moderate statistically significant correlation between BMI and presenting complaints (*p* = 0.035; V = 0.307).

Table 3 shows the results of the correlation between gender, age, and BMI of the respondents and the severity of gastrointestinal complaints after eating selected food groups.

Based on the study, statistical significance was demonstrated between the gender of the respondents and the severity of gastrointestinal complaints after the consumption of legumes and alcohol. The analysis performed confirmed the correlation between the age of the respondents and the severity of complaints when consuming raw vegetables and fruits, brassica vegetables, legumes, fried products, and spicy products. There was also a significant correlation between the body mass index (BMI) of the respondents and the severity of complaints after alcohol consumption.

## 4. Discussion

Functional gastrointestinal disorders involve a heterogeneous group with chronic and recurrent symptoms. In addition, more than 40% of people worldwide struggle with DGBI [5]. Both gender and age, as well as body mass index BMI, may be associated with symptoms and the development of gastrointestinal dysfunction [22]. In the authors’ analysis, in the study population, women made up 70.1% and men 29.9%. Le Pluart et al. [22] also showed female dominance (77.9%; mean age was 49.7 years). In terms of body mass index, 18.6% of the women and 34.8% of the men studied were found to be overweight, while obesity was found among 8.4% of women and 8.6% of men. A larger proportion of respondents (67.4% of women and 55.6% of men) had a normal body weight, comparatively, in our study. Similarly, a large-scale online study by Miwa [23] in Japan on lifestyle among people with functional gastrointestinal disorders showed that functional dyspepsia is more frequently detected in women than among men regardless of age. Similarly, the prevalence of IBS is predominant for women, and this range increases with age.

Functional disorders of the gastrointestinal tract often underlie changes in quality and comfort of life. In addition, they are often a major challenge for physicians when it comes to making an appropriate diagnosis and subsequent therapeutic action [24]. Many times, the change in quality of life is correlated with the presence of chronic and intractable complaints. Among the study group, abdominal pain was the most frequently reported, occurring frequently in 48.3% of people, followed by constipation in 47.4% of respondents, and slightly fewer people suffered from bloating, at 46.8%. On the other hand, 42.6% of people complained of belching, and also 26.9% of respondents complained of excessive gas. Different data were obtained by Zielinska [25], where the most popular complaints reported by respondents included bloating (58%), epigastric pain (54%), constipation (46%), intestinal cramps (46%), and belching (45%). In the case of a study conducted on the American population by Almario CV et al. [26], different results were also shown regarding gastrointestinal symptoms. Heartburn/reflux was present in 30.9% of subjects, followed by abdominal pain among 24.8% of subjects, bloating in 20.6%, diarrhea in 20.2%, and constipation in 19.7%. In contrast, in a study by Bardisi et al. [27], the most common symptoms were excessive gas (22.1%), cramping (19.2%), nausea (15.1%), and vomiting (13.1%).

Symptom exacerbation is significantly related to the type of food provided, such as coffee, fatty and spicy foods, sweeteners, some fruits and vegetables, grain products, and alcohol, among others. In addition, a diet high in carbohydrates and fats can exacerbate symptoms [28]. Göktaş et al. [29] showed that symptom triggers among study subjects with functional dyspepsia, similarly as above, are both fried and fatty foods (27.1%), as well as spicy condiments (21.8%) and carbonated beverages (21.8%). In a study conducted by Hayes et al. [30] among people with IBS, 53.3% of respondents identified foods consisting of cereals, particularly bread, as stimulators of gastrointestinal symptoms. This was followed by 39.3% of respondents who perceived spicy foods and 35.6% of people who perceived vegetables (cabbage, onions, garlic) and foods high in fat (sauces, French fries, fries) as problematic. On the contrary, as in our study, respondents showed that alcohol, raw fruits and vegetables, and brassica vegetables were the least likely to affect gastrointestinal complaints. In contrast, slightly different results were obtained by Bardisi et al. [27], where almost half of the respondents believed that garlic, and 59.9% of people onions, contribute to the appearance of symptoms. In addition, 34.9% also pointed to strawberries. Dairy products, coffee, oats, and cinnamon were also considered troublesome. Lee et al. [31] also showed that milk contributes to causing discomfort (34.7%), in addition to pork belly (24.8%), fast food (23.8%), and high-fat foods and foods containing gluten. Böhn et al. [32] singled out carbohydrate products affecting the appearance of gastrointestinal symptoms. These include dairy products (49%), beans/lentils (36%), and apples (28%). Additionally, fried and fatty foods were found to be mediators of symptom formation. Among the compiled studies, it is possible to distinguish the products most often indicated as stimulators of the formation of complaints. These are mainly dairy products, high-fat, spicy, and fried foods, as well as wheat products and some vegetables (garlic, onions).

DGBI individuals should pay special attention to the supply of sweets and sugary drinks in their diet, since these products are characterized by high levels of FODMAPs. Guo et al. [33] obtained similar results to the presented study: sweets once a week or less often were consumed by the majority of respondents: 64.1% of people with IBS. According to the results obtained in a study by Chiril et al. [34], half of the respondents with functional dyspepsia consumed sweets less than once a week, and the other half at least once a week. In contrast, among those with gastroesophageal reflux, more than half (56.1%) consumed sweets at least once a week.

Another important factor in the diet of people with functional gastrointestinal disorders is limiting alcohol consumption, as some unfavorable correlations have been reported between the consumption of alcoholic beverages and the appearance, or severity, of gastrointestinal discomfort [35]. Huang et al. [36], similar to the authors’ study, showed a low consumption of alcohol: 68 (93.8%) did not consume it. On the other hand, a different relationship was obtained by Chirila et al. [34], as healthy subjects showed a lower range of alcoholic beverage consumption, which is at the level of less than once a week, while 78.6% of respondents with FD and 56.1% with esophageal reflux consumed alcohol more frequently, at least once a week. Also, Nilholm et al. [37] examined the amount of alcohol consumed, where most often, 43.8% of respondents consumed less than one glass of alcohol during the week. 

The diet among the study population of patients varies greatly. Regarding the principles of healthy nutrition, some of the dietary habits and preventions are correct. Slight deviations may be due to the large variety of patients and the different ailments present among the surveyed people. In addition, there are many elements affecting dietary choices. It is not possible to discuss and isolate each of the determinants of dietary behavior among the study group, either in our analysis or in the studies of other authors. The previously cited age, gender, BMI, and education are popular determinants of eating behavior. Therefore, an individualized and multifaceted approach to each DGBI patient is very important. 

### Strengths and Weaknesses of the Study

This study was conducted to evaluate diet and the effect of diet on the exacerbation of symptoms of functional gastrointestinal disorders. One of the strengths of the study is the use of a detailed questionnaire, which takes into account various aspects of diet and functional symptoms of gastrointestinal disorders, and allows a thorough understanding of the relationship between diet and intestinal complaints. The use of recognized diagnostic criteria, such as the Rome IV criteria, increases the reliability of diagnosed disorders.

This survey also has some limitations. However, a weakness of the survey may be the lack of control over all factors that may affect the results, such as environmental variables or other health factors. In addition, the use of only one method of assessing nutritional status, the BMI, may not take into account other important aspects related to diet and lifestyle.

## 5. Conclusions

Analysis of the results revealed the complexity of eating habits in the disorders of gut–brain interaction (DGBI) population. The results identify abdominal pain, bloating, and constipation as the most commonly reported gastrointestinal symptoms among participants. The association between the consumption of certain foods, such as milk, and dairy products, as well as fried and fatty foods, and the severity of DGBI symptoms was confirmed. Despite this, the majority of respondents did not eliminate any food products, to alleviate the discomfort.

The relationship between demographic and anthropometric indicators such as gender, age BMI, and the prevalence and intensity of gastrointestinal symptoms has been confirmed in studies. Also, these relationships relate to the types of foods that exacerbate complaints.

These findings have important implications for clinical dietary practices and may inform the need for a personalized approach in constructing dietary interventions for patients with DGBI. They underscore the need to further study the impact of diet on DGBI and adjust dietary recommendations to minimize symptomatology and improve the quality of life for this group of patients. In addition, these findings suggest the need for extensive dietary education aimed at patients with DGBI, which would include both the elimination of foods with potential symptom-enhancing effects and optimal meal frequency in the context of overweight and obesity prevention.

## Figures and Tables

**Table 1 jcm-13-04132-t001:** Frequency of intake of different food groups among respondents.

Type of Product	Frequency of Consumption					
	Several Times a Day % (*n*)	Once a Day % (*n*)	3 or More Times a Week % (*n*)	Several Times a Month % (*n*)	I Do Not Consume % (*n*)	*p*-Value
Vegetables	37.2% (87)	23.1% (54)	26.9% (63)	12.4% (29)	0.4% (1)	0.034 *
Fruits	39.3% (92)	34.6% (81)	15.0% (35)	10.3% (24)	0.9% (2)	0.003 *
Meat	10.7% (25)	18.8% (44)	26.9% (63)	28.6% (67)	15.0% (35)	0.582
Fish	0% (0)	0% (0)	12.8% (30)	71.8% (168)	15.4% (36)	0.443
Wheat bread	18.8% (44)	23.9% (56)	22.2% (52)	16.7% (39)	18.4% (43)	0.873
Whole-grain bread	17.5% (41)	17.5% (41)	16.2% (38)	14.5% (34)	34.2% (80)	0.043 *
Coarse-grain groats	0.4% (1)	2.6% (6)	19.2% (45)	47.0% (110)	30.8% (72)	0.032 *
Wheat pasta	1.2% (3)	4.4% (10)	35.0% (82)	50.4% (118)	9.0% (21)	0.261
Whole wheat pasta	0.4% (1)	4.4% (10)	15.8% (37)	27.8% (65)	51.6% (121)	0.021 *
Legumes	0% (0)	5.1% (12)	9.8% (23)	61.1% (143)	23.9% (56)	0.041 *
Nonfermented dairy products	20.6% (48)	26.3% (62)	21.1% (49)	4.8% (11)	27.2% (64)	0.231
Fermented dairy products	2.2% (5)	14.9% (35)	32.6% (76)	32.0% (75)	18.3% (43)	0.128
Cottage cheese	1.3% (3)	3.8% (9)	35.4% (83)	20.9% (49)	38.6% (90)	0.123
Rennet cheeses	3.0% (7)	14.2% (33)	46.2% (108)	22.5% (53)	14.2% (33)	0.078
Coffee	44% (103)	20.1% (47)	5.6% (13)	18.4% (43)	12% (28)	0.323
Tea	38.0% (89)	19.2% (45)	13.7% (32)	22.2% (52)	6.8% (16)	0.289
Fast food	0.4% (1)	0% (0)	7.3% (17)	83.3% (195)	9.0% (21)	0.021 *
Sweets	7.3% (17)	0% (0)	35.0% (82)	51.7% (121)	6.0% (14)	0.041 *
Alcohol	0% (0%)	0% (0%)	3.8% (9)	81.2% (190)	15.0% (35)	0.037 *

* = *p* < 0.05.

**Table 2 jcm-13-04132-t002:** Correlation analysis between the occurrence of gastrointestinal ailments and gender, age, and BMI of the respondents.

Category		Nausea	Constipation	Bloating	Belching	Diarrhea	Excessive Passing of Gas	Overflow	Fasting Abdominal Pain	Nighttime Abdominal Pain	Chronic Fatigue and Weakness
Gender X ± SD	Woman *n* = 164	2.55 ± 0.65	1.68 ± 0.76	1.74 ± 0.76	2.18 ± 0.79	2.29 ± 0.79	2.31 ± 0.74	2.40 ± 0.75	2.55 ± 0.77	2.63 ± 0.64	2.30 ± 0.80
	Man *n* = 70	2.67 ± 0.61	1.90 ± 0.87	1.77 ± 0.85	2.14 ± 0.91	2.15 ± 0.88	2.10 ± 0.87	2.14 ± 0.84	2.46 ± 0.86	2.63 ± 0.71	2.57 ± 0.77
	Total *n* = 234	2.59 ± 0.64	1.75 ± 0.80	1.75 ± 0.79	2.17 ± 0.82	2.25 ± 0.82	2.25 ± 0.79	2.32 ± 0.78	2.52 ± 0.80	2.63 ± 0.66	2.38 ± 0.80
	*p*-value	0.179	0.056	0.765	0.774	0.268	0.053	0.020 *	0.422	0.996	0.017 *
Age X ± SD	<30 years *n* = 76	2.55 ± 0.64	1.83 ± 0.77	1.91 ± 0.77	2.25 ± 0.77	2.37 ± 0.73	2.36 ± 0.73	2.41 ± 0.75	2.58 ± 0.75	2.67 ± 0.57	2.14 ± 0.78
	31–40 years *n* = 74	2.51 ± 0.67	1.78 ± 0.78	1.89 ± 0.77	2.16 ± 0.83	2.09 ± 0.83	2.11 ± 0.79	2.23 ± 0.79	2.24 ± 0.90	2.49 ± 0.73	2.31 ± 0.87
	41–50 years *n* = 65	2.69 ± 0.58	1.60 ± 0.83	1.43 ± 0.75	2.15 ± 0.89	2.22 ± 0.89	2.32 ± 0.83	2.35 ± 0.84	2.72 ± 0.65	2.72 ± 0.65	2.83 ± 0.49
	>51 years *n* = 19	2.63 ± 0.69	1.79 ± 0.86	1.63 ± 0.76	1.89 ± 0.81	2.47 ± 0.77	2.16 ± 0.76	2.26 ± 0.73	2.68 ± 0.75	2.68 ± 0.67	2.05 ± 0.85
	Total *n* = 234	2.59 ± 0.64	1.75 ± 0.80	1.75 ± 0.79	2.17 ± 0.82	2.25 ± 0.82	2.25 ± 0.79	2.32 ± 0.78	2.52 ± 0.80	2.63 ± 0.66	2.38 ± 0.80
	*p*-value	0.383	0.360	0.001 *	0.416	0.121	0.205	0.545	0.002 *	0.154	0.00 *
BMI X ± SD	Underweight *n* = 6	2.50 ± 0.55	1.50 ± 0.55	1.67 ± 0.82	1.83 ± 0.75	2.00 ± 0.89	1.67 ± 0.52	2.33 ± 0.82	2.00 ± 1.10	2.00 ± 0.89	1.67 ± 0.52
	Norm *n* = 177	2.55 ± 0.67	1.72 ± 0.79	1.79 ± 0.80	2.19 ± 0.82	2.32 ± 0.79	2.28 ± 0.78	2.30 ± 0.79	2.53 ± 0.79	2.62 ± 0.67	2.37 ± 0.80
	Overweight *n* = 45	2.80 ± 0.46	1.96 ± 0.85	1.56 ± 0.72	2.11 ± 0.86	1.98 ± 0.89	2.20 ± 0.87	2.40 ± 0.78	2.60 ± 0.78	2.80 ± 0.55	2.62 ± 0.72
	Obesity *n* = 6	2.17 ± 0.75	1.33 ± 0.52	2.17 ± 0.41	2.17 ± 0.75	2.33 ± 0.82	2.33 ± 0.52	2.50 ± 0.84	2.33 ± 0.82	2.33 ± 0.52	1.50 ± 0.84
	Total *n* = 234	2.59 ± 0.64	1.75 ± 0.80	1.75 ± 0.79	2.17 ± 0.82	2.25 ± 0.82	2.25 ± 0.79	2.32 ± 0.78	2.52 ± 0.80	2.63 ± 0.66	2.38 ± 0.80
	*p*-value	0.037 *	0.138	0.183	0.719	0.071	0.279	0.828	0.344	0.019 *	0.001 *

X = average; SD = standard deviation; * = *p* < 0.05.

**Table 3 jcm-13-04132-t003:** Analysis of the relationship between gender, age, and BMI of respondents and the severity of gastrointestinal complaints after eating selected food groups.

Category		Milk and Dairy Products	Raw Vegetables and Fruits	Legumes	Coffee	Alcohol	Products High in Fiber	Products High in Fat	Fried Products	Spicy Products
Gender X ± SD	Woman *n* = 164	1.88 ± 0.83	2.36 ± 0.80	2.32 ± 0.80	2.16 ± 0.81	2.27 ± 0.84	2.42 ± 0.81	2.37 ± 0.78	1.91 ± 0.86	1.80 ± 0.86
	Man *n* = 70	1.84 ± 0.71	2.47 ± 0.74	2.44 ± 0.77	2.39 ± 0.79	2.33 ± 0.79	2.73 ± 0.59	2.16 ± 0.77	1.93 ± 0.79	1.76 ± 0.71
	Total *n* = 234	1.87 ± 0.79	2.40 ± 0.79	2.35 ± 0.80	2.23 ± 0.81	2.29 ± 0.82	2.51 ± 0.76	2.30 ± 0.78	1.91 ± 0.84	1.79 ± 0.82
	*p*-value	0.757	0.345	0.269	0.049 *	0.646	0.004 *	0.060	0.868	0.722
Age X ± SD	<30 years *n* = 76	1.95 ± 0.81	2.71 ± 0.63	2.55 ± 0.72	2.46 ± 0.76	2.42 ± 0.80	2.55 ± 0.76	2.39 ± 0.75	2.08 ± 0.84	2.07 ± 0.87
	31–40 years *n* = 74	1.85 ± 0.82	2.27 ± 0.83	2.19 ± 0.87	2.18 ± 0.83	2.16 ± 0.84	2.35 ± 0.83	2.20 ± 0.79	1.89 ± 0.85	1.70 ± 0.75
	41–50 years *n* = 65	1.82 ± 0.75	2.18 ± 0.81	2.31 ± 0.77	2.03 ± 0.79	2.26 ± 0.82	2.65 ± 0.65	2.31 ± 0.79	1.72 ± 0.80	1.57 ± 0.75
	>51 years *n* = 19	1.79 ± 0.79	2.37 ± 0.68	2.37 ± 0.76	2.16 ± 0.83	2.37 ± 0.83	2.53 ± 0.77	2.32 ± 0.82	2.00 ± 0.82	1.74 ± 0.81
	Total *n* = 234	1.87 ± 0.79	2.40 ± 0.78	2.35 ± 0.80	2.23 ± 0.81	2.29 ± 0.82	2.51 ± 0.76	2.30 ± 0.78	1.91 ± 0.84	1.79 ± 0.82
	*p*-value	0.740	0.000 *	0.042 *	0.014 *	0.268	0.135	0.518	0.087	0.002 *
BMI X ± SD	Underweight *n* = 6	1.83 ± 0.98	2.67 ± 0.52	2.33 ± 1.03	2.00 ± 0.89	2.00 ± 1.10	2.00 ± 1.10	2.17 ± 0.98	2.00 ± 1.09	1.83 ± 0.98
	Norm *n* = 177	1.81 ± 0.77	2.44 ± 0.78	2.41 ± 0.76	2.25 ± 0.79	2.32 ± 0.79	2.49 ± 0.77	2.29 ± 0.76	1.94 ± 0.83	1.84 ± 0.83
	Overweight *n* = 45	2.11 ± 0.83	2.18 ± 0.81	2.18 ± 0.89	2.18 ± 0.86	2.18 ± 0.91	2.73 ± 0.58	2.36 ± 0.80	1.80 ± 0.81	1.58 ± 0.72
	Obesity *n* = 6	1.83 ± 0.75	2.50 ± 0.84	2.00 ± 0.89	2.00 ± 1.10	2.50 ± 0.84	2.17 ± 0.98	2.33 ± 1.03	2.00 ± 1.10	1.83 ± 0.98
	Total *n* = 234	1.87 ± 0.79	2.40 ± 0.78	2.35 ± 0.80	2.23 ± 0.81	2.29 ± 0.82	2.51 ± 0.76	2.30 ± 0.78	1.91 ± 0.84	1.79 ± 0.82
	*p*-value	0.154	0.178	0.226	0.730	0.528	0.045 *	0.936	0.780	0.305

X = average; SD = standard deviation; * = *p* < 0.05.

## Data Availability

The data presented in this study are available on request from the corresponding author. The data are not publicly available due to privacy.

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
