# Peer review of "Evaluation of Diet and Symptom Severity in Disorder of Gut–Brain Interaction"

_jcm, 2024, doi:10.3390/jcm13144132_

Round 1

Reviewer 1 Report

Comments and Suggestions for Authors

In this manuscript the authors analyzed diet and symptom severity in disorder of Gut-Brain Interaction.

The topic of the manuscript is very interesting, but still there are certain shortcomings:

ü  Introduction, lines 58-69: the proposal is to remove this pause from the text of the manuscript because the narration deviates from the main goal of the work. It is enough to briefly highlight the Roma Criteria

ü  it is necessary to analyze in more detail the pathogenesis of disorders caused by inadequate gut-brain interaction

ü  drop Table 1

ü  the methodology was done precisely, with a detailed analysis of all aspects

ü  the results of the manuscript are mostly based on descriptive statistics and the chi-square test, with a lot of overlapping of the text part with the content listed in the tables

ü  in this sense, it is necessary to edit Tables 3 and 4 because they are very unreadable

ü  in order to obtain a greater practical value of the results, it is necessary to perform regression analysis instead (in addition to correlation); thus, concrete predictors of interaction gut-brain disorders would be obtained

ü  avoid repeating the description of the obtained results in the part of the manuscript that refers to the discussion

Author Response

Thank you so much for taking the time to evaluate our work. We have tried to incorporate all your valuable suggestions. If we could improve our work in any way, please let us know.

Comment 1.

 Introduction, lines 58-69: the proposal is to remove this pause from the text of the manuscript because the narration deviates from the main goal of the work. It is enough to briefly highlight the Roma Criteria.

Thank you for your guidance. Changes have been made as recommended.

Comment 2.

It is necessary to analyze in more detail the pathogenesis of disorders caused by inadequate gut-brain interaction.

Thank you, we elaborated on this in the introduction of the study.

Comment 3.

Drop Table 1.

Thank you for your guidance. Changes have been made as recommended.

Comment 4.

The methodology was done precisely, with a detailed analysis of all aspects.

Thank you very sincerely.

Comment 5.

The results of the manuscript are mostly based on descriptive statistics and the chi-square test, with a lot of overlapping of the text part with the content listed in the tables.

Thank you for your suggestion, we have made every effort to improve the manuscript.

Comment 6.

In this sense, it is necessary to edit Tables 3 and 4 because they are very unreadable.

The tables have been edited.

Comment 7.

In order to obtain a greater practical value of the results, it is necessary to perform regression analysis instead (in addition to correlation); thus, concrete predictors of interaction gut-brain disorders would be obtained.

Please indicate specifically what elements we should test. Thank you for your help.

Comment 8.

Avoid repeating the description of the obtained results in the part of the manuscript that refers to the discussion

.

Thank you for your guidance. Changes have been made to the discussion.

Thank you for your help. Your guidance is invaluable.

Kind regards,

Authors.

Reviewer 2 Report

Comments and Suggestions for Authors

The authors of “Evaluation of Diet and Symptom Severity in Disorder of Gut Brain Interaction” perform an interesting work due to the gut-brain axis being a hot topic in the microbiota field. After reading the article, is clear that the uthors performs a great clinical work, however some topics need attention and revision.

First, is not clear how the authors evaluate the impairment on the Gut-brain interaction regarding the diet consumption. The symptoms detected are related to typical disorders in the gastrointestinal tract that could be related to microbiota disorders. I recommend changing the title of the work, and focusing on the gut effects registered, but not the gut-brain interaction, which could be confusing for readers interested in gut-brain axis works. The effect of gut-brain interaction is a possible explanation of the observed symptoms, but is not clear to me that this work evaluated gut-brain interaction through a survey. Also, no gut microbiome were analyzed in the patients. Also, the survey used should be shared in annexes as supplementary information.

Correct p<0.05. The statistical correct term is p-value.

Correct Chi2. Use the statistical correct term.

Author Response

Thank you so much for taking the time to evaluate our work. We have tried to incorporate all your valuable suggestions. If we could improve our work in any way, please let us know.

Comment 1.

First, is not clear how the authors evaluate the impairment on the Gut-brain interaction regarding the diet consumption. The symptoms detected are related to typical disorders in the gastrointestinal tract that could be related to microbiota disorders. I recommend changing the title of the work, and focusing on the gut effects registered, but not the gut-brain interaction, which could be confusing for readers interested in gut-brain axis works. The effect of gut-brain interaction is a possible explanation of the observed symptoms, but is not clear to me that this work evaluated gut-brain interaction through a survey. Also, no gut microbiome were analyzed in the patients. Also, the survey used should be shared in annexes as supplementary information.

Thank you for your guidance.

The title of the study, "Evaluation of Diet and Symptom Severity in Disorder of Gut-Brain Interaction," includes the term "Disorder of Gut-Brain Interaction" because this terminology reflects the updated nomenclature that replaced "Functional Gastrointestinal Disorders." The new terminology, introduced with the Rome IV criteria, aims to better capture the complex, bidirectional interactions between the gut and the brain that contribute to these disorders. This modernized term highlights the multifactorial nature of these conditions and aligns with the latest understanding in the field.

The survey has been included as supplementary material.

Comment 2.

Correct p<0.05. The statistical correct term is p-value.

Thank you for your guidance. The changes have been made in the manuscript.

Comment 3.

Correct Chi2. Use the statistical correct term.

Thank you for your guidance. The changes have been made in the manuscript.

Thank you for your help. Your guidance is invaluable.

Kind regards,

Authors.

Reviewer 3 Report

Comments and Suggestions for Authors

Functional GI disorders represent a high incidence all over the world. Hence, the gastroenterologist clinics are flooded with such patients (around 75%) of all gastroenterologists OPD patients. The chronicity of such conditions represents pain in the neck for both the patients themselves and their doctors. The quality of life of the patients is affected too much. The main issue for gastroenterologists is to exclude organic causes of the complaints. This entails a lot of tests including blood tests and radiological images with high costs and pressure on the medical facilities. Once diagnosed with a functional disease, the doctors explain the condition to their patients and try to make all efforts to make the patients lead their lives in a way that improves their symptoms i.e. dietary personalized approach. Being, multifactorial, dietary manipulation is not always of help.

The authors try to shed light on diet manipulation's effect on Functional GI diseases. They do not add any more new data to the known literature.

I have a few comments:

Line 117: A survey questionnaire was used to conduct the study----> you should attach the questionnaire sheet.

Line 161: Inclusion criteria--------> It needs to be rephrased. You can say all patients diagnosed with gastrointestinal dysfunction in the gastroenterologist clinic who would like to participate in the study and correctly complete the questionnaire.

Line 165: Exclusion criteria:  the presence of food allergies or intolerances------------------------>How you can diagnose this disease. It is better to say the history of.....

Line 241:  up to 1/4 of the population of 241 Western countries struggles with DGBI-------->The DGBI is a cosmopolitan condition and it is better to give an international figure as that in line 48 "more than 40% of people worldwide suffer from DGBI".

Line 331: Strengths and weaknesses of the study------> Being diagnosed by a gastroenterologist is not a strength point!!! and even the number {234} and gender...!!!!

Line 343: the lack of a control group------> such studies could not be controlled!!

Author Response

Thank you so much for taking the time to evaluate our work. We have tried to incorporate all your valuable suggestions. If we could improve our work in any way, please let us know.

Comment 1.

Line 117: A survey questionnaire was used to conduct the study----> you should attach the questionnaire sheet.

Thank you for your guidance. The survey has been included as supplementary material.

Comment 2.

Line 161: Inclusion criteria--------> It needs to be rephrased. You can say all patients diagnosed with gastrointestinal dysfunction in the gastroenterologist clinic who would like to participate in the study and correctly complete the questionnaire.

Corrected in the text according to the comment.

Comment 3.

Line 165: Exclusion criteria:  the presence of food allergies or intolerances------------------------>How you can diagnose this disease. It is better to say the history of.....

The exclusion criterion "diagnosed food allergies or intolerances" was established to avoid influencing the study results. Food allergies and intolerances can significantly alter gastrointestinal symptoms, potentially distorting the data collected. These conditions had to be confirmed by the attending physician. We did not interfere with the method of diagnosis, as we trusted that the specialist had the appropriate qualifications and expertise to make an accurate assessment.

Comment 4.

Line 241:  up to 1/4 of the population of 241 Western countries struggles with DGBI-------->The DGBI is a cosmopolitan condition and it is better to give an international figure as that in line 48 "more than 40% of people worldwide suffer from DGBI".

Thank you for your guidance. Changes have been made to the text as recommended.

Comment 5.

Line 331: Strengths and weaknesses of the study------> Being diagnosed by a gastroenterologist is not a strength point!!! and even the number {234} and gender...!!!!

We have corrected the indicated passage, however, if you think it should be changed or information added to it please indicate to us specifically what your requirements are. Thank you.

Comment 6.

Line 343: the lack of a control group------> such studies could not be controlled!!

Thank you for your guidance. Changes have been made to the text as recommended.

Thank you for your help. Your guidance is invaluable.

Kind regards,

Authors.

Round 2

Reviewer 1 Report

Comments and Suggestions for Authors

I thank the authors for the answer. Basically, most of the suggestions are accepted, it remains to do a regression analysis of the influence of gender, age and BMI on the severity of gastrointestinal problems. This would give the obtained results a special, predictive significance

Author Response

Dear Reviewer,  Thank you very much for your suggestion. We have added a relevant section in the results describing the correlation between the variables. We have also added a description of the tests used in the survey methodology.   Thank you.
